


# An automatic DI-flux at the Livingston Island Geomagnetic Observatory, Antarctica: requirements and lessons learnt

Santiago Marsal[1], Juan José Curto[1], Joan Miquel Torta[1], Alexandre Gonsette[2], Vicent Favà[3], Jean Rasson[2], Miquel Ibañez[1], Òscar Cid[1]

[1]Observatori de l'Ebre, (OE), CSIC – Universitat Ramon Llull, 43520 Roquetes, Spain.
[2]Institute Royal Météorologique (IRM), Centre de Physique du Globe, 5670 Viroinval (Dourbes), Belgium.
[3]Institut de l'Ebre, 43500 Tortosa, Spain.

*Correspondence to*: Santiago Marsal (smarsal@obsebre.es)

**Abstract.** The DI-flux, consisting of a fluxgate magnetometer bar coupled with a theodolite, is used for the absolute manual

measurement of the magnetic field angles in most ground-based observatories world-wide. Commercial solutions for an automated DI-flux have recently been developed by the Royal Meteorological Institute of Belgium (RMI), and are practically reduced to the AutoDIF and its variant, the GyroDIF. In this article, we analyse the pros and cons of both instruments in terms of its suitability for installation at the partially manned geomagnetic observatory of Livingston Island, Antarctica. We conclude that the GyroDIF, even if less accurate and more power demanding, is more suitable than the

AutoDIF for harsh conditions due to its simpler necessary infrastructure. Power constraints in the Spanish Antarctic Station during the unmanned season impose an energy-efficient design of the thermally regulated box housing the instrument, as well as a thorough power management. Our experiences can benefit the geomagnetic community, who often faces similar challenges.

## 1 Introduction

Ground-based geomagnetic field data are currently used in a number of scientific works, from the Earth's deep interior to Space Weather studies, the latter with clear practical implications on our current society. As a consequence of these facts, the developments in instrumentation, data acquisition and data dissemination have increased the interest of the scientific community, and one of the most challenging aspects is the augmentation of the data coverage on remote sites such as oceanic regions in general, and the Southern Hemisphere in particular. For both logistical and economic reasons, full

automation of the data acquisition is desirable, especially at those remote sites. There are several elements of geomagnetic observatory operation which have been identified to be fully or partially automated: data collection, data telemetry, data processing, data dissemination, error detection and absolute observations (Newitt, 2007). This paper is aimed at shedding some light on the practical aspects of the automation of absolute observations. We will report on the lessons learnt from the installation of an automatic absolute magnetometer in a particularly adverse environment, as it is our partially manned

station in Antarctica.





The Livingston Island geomagnetic observatory (62.7° S, 60.4° W, coded as LIV by the International Association of Geomagnetism and Aeronomy) is located in the Spanish Antarctic Station Juan Carlos I (ASJI), in the South Shetlands, north of the Antarctic Peninsula. Its first installation took place during the 1995–1996 and 1996–1997 Antarctic Surveys and it has

records since December 1996. This observatory is manned during the austral summer months, typically from the end of November to February, being in automatic operation without human intervention the rest of the year. In terms of measuring instruments, it currently has three variometric magnetometers: a proton vector magnetometer in dIdD configuration, an FGE triaxial fluxgate magnetometer, and a GEM Systems scalar magnetometer. As for absolute instruments, it has a DI-flux consisting of a Carl Zeiss THEO 015B theodolite equipped with an 810 Elsec Fluxgate probe, and another GEM Systems

proton magnetometer. For a detailed description of these instruments and the utility of the data provided by them see Pijoan et al. (2014) and references therein. For the purposes of this paper, it is interesting to note that variometers in general are automatic instruments with relatively high resolution, especially the combination FGE-scalar magnetometer, but they do not rely on a stable reference frame. The DI-flux, on the contrary, is based on a fixed reference frame, but its measurements have a lower resolution and, most importantly, they are necessarily manual. Thus, lacking absolute measurements during nine

months a year when the station is unmanned prevents us from establishing reliable baselines for reduction of our variometer data. Since we don't know what the baseline evolution is during those nine months, we currently assume a simple linear variation between consecutive surveys.

During the last 20 years we have progressed in practically all of the aforementioned aspects concerning the automation of

LIV geomagnetic observatory. However, as for any institute that runs remote magnetic observatories, the automation of the absolute observations is of particular importance and the most challenging item, especially when the station is unmanned most of the time. At present, there have been very few attempts to automate absolute observations (Hrvoic and Newitt, 2011). The one with longer history is called AutoDIF (Rasson and Gonsette, 2011; Gonsette and Rasson, 2013), an automated DI-flux basically designed to reproduce its manual measurement sequence. For what concerns the Declination

measurement, the telescope of its theodolite is replaced by a laser and split photo cells which are used to align the device in a known meridian by reflecting the laser beam off a corner cube reflector back onto the photo cell. A recent variant is being provided that substitutes the target pointing system by an embedded device by which the True North referencing is achieved by a rate-gyroscope able to detect the Earth rotation. This variant is accordingly called GyroDIF. Both AutoDIF and GyroDIF use non-magnetic piezoelectric motors to move the sensor about the horizontal and vertical axes. The angles are

measured by custom electronic optical encoders. An electronic bubble level mounted on the alidade provides reference to the horizontal. An embedded fan-less PC and a microcontroller govern the instrument. In-house testing has shown that the instrument can achieve an angular accuracy of 0.1′ (arc-minute), which is comparable with the one that can be obtained by a skilled observer with a DI-flux. Although this instrument has given results that agree closely with those obtained by manual observations, long-term reliability under adverse conditions must yet be demonstrated (Hrvoic and Newitt, 2011).



In this article, we show our recent experience leading to the installation of a GyroDIF at the ASJI in January–February 2017. This comprises the choice of the most suitable automatic absolute instrument based on the particular conditions in our station, as well as the design of the necessary infrastructure to accommodate it. Because the instrument deployment is still in progress our experience is limited, so in this text we will combine in situ LIV data with real data taken in Ebre Observatory

headquarters during a period of test in 2016.

## 2. AutoDIF vs. GyroDIF

First, we had to assess the most suitable option between the two above: the AutoDIF or the GyroDIF. Assuming mechanical stability of the pillars where the variometers are deployed, just a few absolute magnetic determinations per week are required. The uncertainty of a single declination measurement with the AutoDIF is typically below 0.3′ if the laser reflector

is far enough, which is translated into less than 2 nT in the magnetic east component ($Y$) at LIV. On the other hand, the north-seeking gyroscope procedure of the GyroDIF, which is used for reference in the Declination measurements, has an expected standard deviation $\sigma_0$ around 3.6′, which translates into an uncertainty of 19 nT in $Y$ at LIV. Fortunately, however, the dispersion of the latter measurements shows a white noise signature, allowing it to be overcome by a sufficiently large number of them. In continuous operation, the GyroDIF uncertainty can thus be substantially reduced.

Another factor to be considered for our choice was power consumption, since the ASJI relies on wind generators during winter. The total average power (including idle and active periods) required for the AutoDIF plus the power needed to keep the instrument above 5 ºC (its minimum working temperature) is probably less than 15–20 W. For the GyroDIF operating in a continuous mode, however, we need a constant temperature, demanding below about 15 W for the heating if we are

capable of providing a good thermal insulation. Adding the power for the instrumentation, as well as that for management of the station, this value raises to more than 70 W.

Thirdly, we had to assess the necessary infrastructure for each kind of equipment. The AutoDIF requires a clear path for the laser beam between the instrument and the reflector target which is difficult to get in the Antarctic environment because of

weather conditions resulting in reduced visibility (snow, fog, rain …). The reflector needs to be separated by at least 30 m from the instrument, though preferably 100 m. A first possibility would be to underground the installation. However, the difficult terrain and the fact that Livingston Island is a protected environment hamper this option. The second possibility was to build a pipe visually connecting the AutoDIF and the reflector, but the strong winds and the instability of the terrain again entailed technical difficulties. The GyroDIF option, on the other hand, is simpler: we just needed to provide a highly

insulated box containing the instrument. Therefore, even if less accurate and more power demanding, the GyroDIF seemed the best option concerning both logistics and stability, and we finally opted for it.



Table 1 summarizes the main differences between the AutoDIF and the GyroDIF from the point of view of its suitability at the target site.

## 3. Installation requirements

Let us analyze the GyroDIF specific requirements in more detail:

- To avoid damage in the piezo-motors acting on the horizontal and vertical axes, the instrument must work above 5 ºC.

- We need to maximize the gyroscope True North sampling so as to reduce the random uncertainty of the Declination observations.

- The gyro response and, in consequence, the uncertainty in the magnetic $Y$ component, also depends on the temperature variation during a single measurement, so thermal stability must be guaranteed.

- The gyroscope response is also sensitive to external accelerations; in consequence, the instrument location requires no motion by wind, sea waves or others, which introduce additional noise into the True North measurements.

- Finally, it is essential to minimize power consumption, preserving it in the system batteries in prevention of periods of scarce wind generation.

### 3.1 Thermal model of the GyroDIF box

To achieve the aforementioned requirements we designed a thermally insulated box accommodating the GyroDIF. Thermal stability within the enclosure is guaranteed by a regulated heating system based on a resistant cable of the type commonly used in underfloor heating installations in buildings. The fundamental idea is to store the heat released by the radiating cable in masonry blocks having the largest available specific heat capacity. Combining the blocks with a high thermal insulation provides the needed thermal stability for the optimum performance of the built-in gyroscope.


The entire box is located within a fiberglass dome or 'igloo' (Fig. 1), which constitutes a first barrier to the external weather conditions. The floor and the walls of the inside box, 25 cm thick, are made of rigid polyurethane foam (PUR) (Fig. 2), while additional foam glass insulation, highly resistant to compressive strength, has been put on the top of the pillar holding the instrument for insulation purposes. PUR is also used to wrap the pillar. Inside the box, a layer of dense bricks, a layer of sand

containing the heating resistance, and an additional batch of bricks around the instrument are aimed at providing the required thermal momentum. The heating system is made up of a floor-integrated 180 W electrical resistance arranged in three height levels, spanning a total length of 9 m. Temperature regulation is achieved by a proportional-integral-derivative (PID) controller switching on and off the electrical resistance on demand, so that the time-integrated electrical power released into the box is balanced by the thermal losses imposed by the outdoor weather conditions.


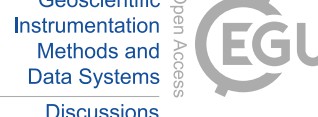



Assuming that the heat is basically lost by conduction through the walls and floor, the total heat power loss of the box can be approximated by:

$$P_l = \lambda_w \frac{S_w}{d_w} \Delta T_w^{io} + \lambda_g \frac{S_g}{d_g} \Delta T_g^{io} + P_a \,, \tag{1}$$

where the $\Delta T^{io}$ terms are the differences between the indoor and outdoor temperatures; $\lambda$, $S$ and $d$ are the thermal
conductivity, the effective surface and the thickness of the insulating material of the box, respectively, and the subscripts $w$ and $g$ in the previous variables stand for 'walls' and 'ground', respectively. Thus, the first and second terms on the right hand side of (1) can be identified with the heat being lost through the walls and ground. The term $P_a$ includes additional losses which are difficult to evaluate a priori (e.g., junctions of the insulating parts, cable conduits entering the box, effect of the pillar where the instrument is supported …), though this term is also expected to be proportional to the temperature gap.

It follows from (1) that the losses are reduced when a minimum gap $\Delta T^{io}$ between the indoor and the outdoor temperatures exists; however, the 5 ºC constraint specified by the manufacturer imposes an optimal working temperature around 7 ºC. Because the ASJI is located near the sea in the North of the Antarctic Peninsula, the average winter temperature is mild: about -6 ºC, with typical variations of $\pm 6$ ºC, while that in summer is $2 \pm 4$ ºC. Temperature drops below -15 ºC are not rare, though the associated weather conditions use to persist for no longer than a few days. $\Delta T_w^{io}$ in winter is thus around 13 ºC on
average, while $\Delta T_g^{io}$ is a few degrees less. Given the size of the box, and assuming $\lambda_w = \lambda_g = 0.027$ W m$^{-1}$ ºC$^{-1}$, $P_a = 4$ W, we get $P_l = 12 \pm 5$ W in winter ($5 \pm 3$ W in summer), which compares well with the experimental values deduced from the electrical power being released within the enclosure.

Because the gyro output is at least affected by a temperature-dependent bias (e.g., Rasson and Gonsette, 2016), it is
important to keep a constant temperature inside the box; thus, the second key point consists in achieving the maximum thermal momentum. This is attained in practice by the masonry blocks located within the box, and can be monitored as the cooling rate after reaching a certain temperature. The (internal) temperature ($T^i$) decay over time ($t$) within the box is approximated by the following formula:

$$-\frac{dT^i}{dt} = \frac{P_l}{C} \,, \tag{2}$$

so that the cooling rate is proportional to the heat power loss ($P_l$) and inversely proportional to the heat capacity of the system ($C$).

Figure 3 shows the temperature evolution during one day of test at Ebre Observatory, prior to heating the box. Green, blue and red lines show the outdoor, the igloo and the box temperatures, respectively. It can be seen that the igloo roughly filters
half of the thermal cycle, while the temperature in the box is drastically reduced to a few percent of the outdoor signal. The figure is aimed at showing the effectiveness of the insulation. Note that the diurnal thermal cycle is virtually inexistent at LIV, especially in winter. Weather fronts, however, with typical periods of a few days, are expected to enter the box, though



with a significant attenuation. The residual temperature variation, nevertheless, is compensated by the PID control (see Sect. 3.2).

As expected from Eq. (1) and (2), and assuming a constant external temperature equal to the seasonal average, the time
response of the temperature decay after heating the box is roughly exponential, with a time constant $\tau$ that is characteristic of the system and can be estimated as the value of $\Delta T^{io} C / P_l$ at any given time, including the initial state. Assuming a specific heat capacity $C_{S,bl} = 800$ J kg$^{-1}$ ºC$^{-1}$ and a mass of 300 kg for the batch of blocks; a specific heat capacity $C_{S,PUR} = 1500$ J kg$^{-1}$ ºC$^{-1}$ and a mass of 100 kg for the PUR, we get $C = 3.9{\times}10^5$ J ºC$^{-1}$. With the above stated values for $P_l$, $C$ and $\Delta T^{io}$, we get estimated cooling rates of 0.11 ˚C h$^{-1}$ in winter and 0.05 ˚C h$^{-1}$ in summer, implying a characteristic time constant $\tau$
between 4 and 5 days. Figure 4 shows experimental temperature decay after heating the box during tests at Ebre Observatory. The empirical time constant is $\tau = 4.6$ days (= 110 h; see exponent of the inset equation in the figure), which is consistent with the estimated value above.

### 3.2 PID thermal control of the box

A simple on-off control scheme was initially tested to achieve temperature control within the GyroDIF enclosure by means
of a heating resistance, but due to the relative high power of the pulses injected into the heater and the thermal momentum of the system, we observed an oscillation with a rate of change at the limit of the maximum recommended by the GyroDIF manufacturer (about 0.1 °C h$^{-1}$). Removing this oscillation as much as possible has been the main reason for implementing a PID control. This type of controller consists of a combination of proportional, integral and derivative control (Fig. 5a). The proportional action generates an output obtained by multiplying the error $e$ (difference between the set point and the real
temperature at the output) by the constant $K_p$. The higher the value of $K_p$, the lower the steady-state error, but in contrast, the system will become more unstable, generating longer transients and oscillations of greater amplitude. As this type of control cannot completely remove the steady-state error, it is combined with an integral control. The integral action generates a value at its output that is obtained by multiplying the time integral of the error $e$ by the constant $K_i$. It has therefore a memory effect, in the sense that the output generated depends on the accumulation of the previous errors rather than on the
current error. This allows the controller's output to achieve a null steady error. However, the integral action, as the proportional action, tends to generate oscillations (especially for increasing $K_i$ values), which can be attenuated by means of a derivative control. The derivative action generates an output proportional to the derivative of the error multiplied by the constant $K_d$. This allows injecting more thermal power into the system when the error is rising, and vice versa, resulting in an attenuation of the oscillations introduced by the proportional and integral actions. This type of control is very vulnerable
to noise, so we adequately filtered the signal from the temperature sensor. To sum up: any perturbation in the system is instantaneously balanced by the derivative action in the sense of minimizing its effects, any difference between the set point

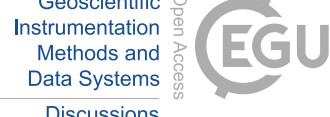



and the current output is corrected by the proportional action, while in a steady state, the control action comes from the integral part.

Several tests were performed in order to evaluate the thermal characteristics of the system and, from them, we could choose the best PID parameters to conciliate a stable temperature in the steady state phase with a quick heating in the transient phase (rise time) with minimum overshoot in the settling time. The final values of the parameters were: $K_p = 50$, $K_i = 0.1$, and $K_d = 300$. Figure 5b shows the effect of the PID control tuned with the former parameters on the GyroDIF box. The temperature is observed to rise from the outdoor temperature up to the working one in about one day (transient state). In the steady state, there is a diurnal thermal oscillation of about 0.3 ºC, with a maximum variation rate of 0.03 °C h$^{-1}$. It should be noted,

however, that this oscillation is not generated by the control system itself, but is a consequence of the external diurnal oscillation (about 15 °C day$^{-1}$ in Ebre Observatory headquarters, where the tests were carried out). This oscillation is much weaker at Livingston Island because of its maritime climate. Preliminary tests at LIV show a diurnal oscillation amplitude within the box circa 0.1 ºC in the steady state.

We note that the thermal control operates in an asymmetric way in our case, because we can actively introduce heat into the box with an electric current through a resistance, but the opposite is left to the natural cooling.

**3.3 Maximum number of gyroscope measurements**

The set of the north-seeking gyro measurements is treated in a similar fashion as regular magnetic measurements in that a baseline is adjusted to observations, except that the True North direction does not change over time as magnetic North does.

Thus, given the substantial random uncertainty of the individual determinations (around $\sigma_0 = 3.6'$; see Sect. 2), there is need to either filter or fit the observations to a known function so as to allocate a single value of the True North reference to each magnetic absolute measurement. We have currently opted for a Gaussian filter, i.e., the convolution of a Gaussian function with the True North observations, so that the maximum weight for a magnetic absolute determination at time $t_0$ is conferred to contemporary gyro measurements, while it is gradually reduced as the time shift increases.


The total time width of the Gaussian filter, $2\sigma$, must be selected adequately. On the one hand, because a single gyro sequence lasts 2 h, a small $\sigma$ value would reduce the amount of available gyro measurements, thus preventing a significant reduction of the statistical (random) error. On the other hand, too large a width would cause the origin of the GyroDIF horizontal angles to drift substantially with respect to True North during that interval (e.g., due to pillar tilting); in other

words, it would filter realistic frequencies of oscillation. For a Gaussian filter with a given $\sigma$, the cut-off period is normally taken at $T_c = \frac{2\pi\sigma}{\sqrt{\ln 2}}$ (half power point or 3 dB attenuation), while the uncertainty associated with the filtered data is given by $\sigma_f = \frac{\sigma_0}{\pi^{1/4}\sqrt{N}}$, where $N$ is the number of measurements in the time interval $2\sigma$. Thus, considering a total width interval ($2\sigma$)





of 3 days on the basis of uninterrupted gyro measurements of the True North allows reproducing typical periods of the pillar drift above 11 days, while it reduces the random uncertainty down to $\sigma_f = 0.5'$. This renders below about 3 nT uncertainty in $Y$ at LIV, thus fulfilling the 5 nT accuracy standard for definitive data required by the INTErnational Real-Time MAGnetic observatory NETwork (INTERMAGNET).

Figure 6 shows a series of gyro measurements in terms of the trace, which is the azimuth (angle from True North) of the 0º reference of the horizontal circle of the GyroDIF theodolite. Superposed are the adopted baseline, the experimental $\sigma_0$ and $\sigma_t$ uncertainties (note they are close to the above estimated values) and the 3-day width Gaussian filter centred in the middle of the measurement interval. The observations were carried out at Dourbes observatory during a previous test period.

**4 Control system electronics**

The control of our new station is based on an Arduino PC. Arduino technology provides a high versatility and low consumption, both characteristics being very convenient for our aim. This control monitors the state of the key elements by means of a series of temperature, current and voltage probes (see Fig. 7). According to these measurements, the control evaluates the power availability in the mains and the charge in the batteries (BAT1 and BAT2), as well as the temperature
conditions in the GyroDIF enclosure (Fig. 7). With this information, the control decides whether or not to feed the different parts of the system by opening and closing solid state switches (r1 to r4 in Fig. 7). After long periods without wind, the batteries diminish the charge feeding the system. When the charge goes under a prefixed threshold and before it can cause irreversible damage to the batteries, the Arduino orders a shut down to the PC that controls the GyroDIF, it turns the heating system off and remains in a standby state awaiting the charge conditions to recover. The same occurs when the suitable
temperature in the GyroDIF enclosure cannot be kept. The Arduino software has an implemented PID algorithm (see Sect. 3.2) controlling the heating power by regularly turning on and off an electric resistance in a pulsed mode, so that the active time in each cycle is proportional to the power delivered to the inertial thermal mass. A second Arduino supervises the whole system and takes the control if the first Arduino fails in its task. A watchdog was implemented to secure an automatic reset in case of a software malfunction, so as to recover the control when the system works unmanned and the staff cannot easily
access the equipment to reboot it.

**5 Power availability**

During winter, power availability is an issue of concern at the ASJI. The consumption of the past instrumentation working during the unmanned period in the station was about 80 W, which includes the three variometres cited in the introduction (FGE, dIdD and Proton) and a satellite transmission system. The consumption of the new GyroDIF system comprises the
following aspects:

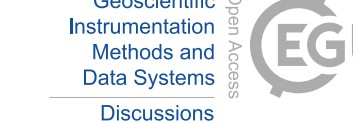



- Instrumentation, comprising the GyroDIF theodolite, its electronic console and control PC: 50 W.
- GyroDIF box heating system: $12 \pm 5$ W (winter average).
- Power management, comprising control, conversion and storage: 13 W.

This implies an additional consumption of $75 \pm 5$ W ($69 \pm 3$ W in summer). Given the wind power production (9 kW) and the effective storing capability (30 kWh) of the station, we estimate an autonomy around 8 days, which is roughly the average maximum time interval without wind at the ASJI.

**6 Summary and conclusion**

Until now, reliable baselines in the Livingston Island geomagnetic Observatory were limited to three months per year
(typically December through February), when the ASJI is operated. The new GyroDIF instrument is expected to provide an uninterrupted series of absolute measurements to reduce the magnetic variations. To this aim, we must firstly guarantee a continuous and reliable power supply providing about 150 W to our magnetic station, which is feasible with the augmentation of the alternative power system that is planned for the next summer survey. We also need this system to be perdurable, which implies continuous renovation of the battery bank and accurate maintenance of the wind generators from
the base.

Secondly, for the proper performance of the integrated optical gyroscope, we need to provide thermal stability to the instrument, implying good insulation and high thermal momentum. This has been achieved with an insulated, thermally regulated enclosure for the GyroDIF, providing slow steady state temperature variations below 0.03 ° h$^{-1}$.

The number of gyroscope measurements is critical for an appropriate characterization of the True-North baseline, which is essential for the correct determination of Declination. The number of D and I measurements themselves is not so critical, and it could be reduced to just a few per week. The uncertainties in the final magnetic field components will comply with the INTERMAGNET conditions if uninterrupted power supply is provided. The less accurate component is $Y$ (East), with an
expected uncertainty amounting to less than 3 nT during the unmanned season.

A robust electronic system, which is duplicated in some parts, has been designed to face the adverse conditions of the winter season, when the ASJI is unmanned. The intelligent Arduino-based control manages the passage of current through the different parts of the system in terms of power availability, and it integrates a PID algorithm adjusting the temperature of the
GyroDIF box.



The necessary infrastructure of the new GyroDIF system has been successfully installed during the last austral summer survey, i.e., between December 2016 and February 2017. The instrument itself, however, is currently (as of February 2017) under installation, and it is expected to be left running unmanned during the next austral winter seasons (March–December).

**Data availability**

No datasets have been used in the production of this article, except those experimental data obtained by the authors to generate figures 3, 4, 5b and 6.

**Author contribution**

S. Marsal designed part of the infrastructure containing the instrument, helped assembling and installing it in Antarctica, leads the data processing, and prepared the manuscript with contributions from all co-authors. J. J. Curto helped designing
part of the infrastructure and electronics. J. M. Torta and J. J. Curto are the Principal Investigators of this project. A. Gonsette and J. Rasson are the instrument manufacturers. M. Ibañez, V. Favà and O. Cid helped designing the electronics controlling the instrument. M. Ibañez has leaded the installation of the necessary infrastructure in Antarctica.

**Competing interests**

The authors declare that they have no conflict of interest.

**Acknowledgments**

The presented results are part of the project CTM2014-52182-C3-1-P funded by the Spanish Ministerio de Economía y Competitividad. We are also grateful to the project 2016-URL-Trac-001 of the Universitat Ramon Llull. SM acknowledges the support of Universitat Ramon Llull through project 2016-URL-IR2Q-029 funded by "Obra Social la Caixa".

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





**Tables:**

|  | AutoDIF | GyroDIF |
|---|---|---|
| $D$ ($Y$) component uncertainty | $< 0.3'$ ($\delta Y < 2$ nT at LIV) <br> limited by the laser pointing procedure | 1 gyro sequence $\Rightarrow 3.6'$ ($\delta Y \approx 19$ nT). <br> Quasi-continuous mode: $\delta Y \lesssim 3$ nT |
| Power consumption | $\approx 20$ W in average (total). | Heating: $\approx 12$ W (average). <br> Instrumentation and energy management: $\approx 60$ W (average). <br> Total: $> 70$ W |
| Necessary infrastructure | Complex: pipes or buried infrastructure. | Simple: thermally insulated box. |

Table 1. Comparative table AutoDIF vs. GyroDIF in terms of suitability at the ASJI.



**Figures:**

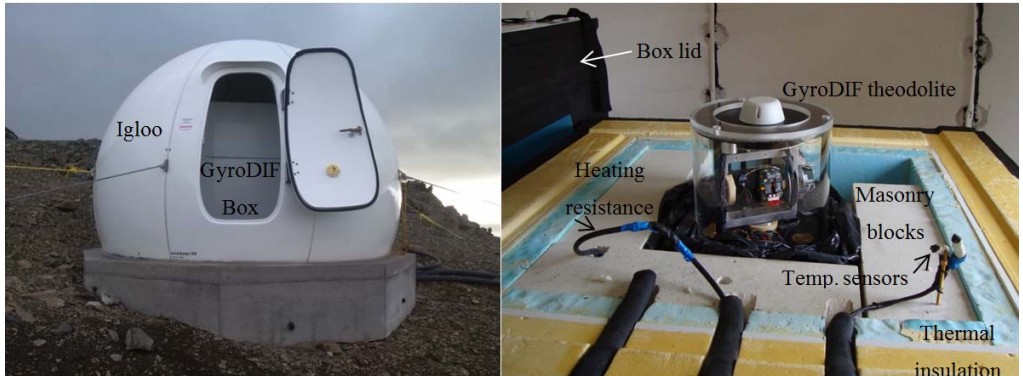

Figure 1: Left: Fiberglass igloo and insulating box (interior) for the GyroDIF thermal insulation at LIV. Right: GyroDIF box with
lid open inside the igloo; the GyroDIF theodolite is visible in the centre, along with the insulation, the masonry blocks, the
temperature sensors and the end of the heating cable.

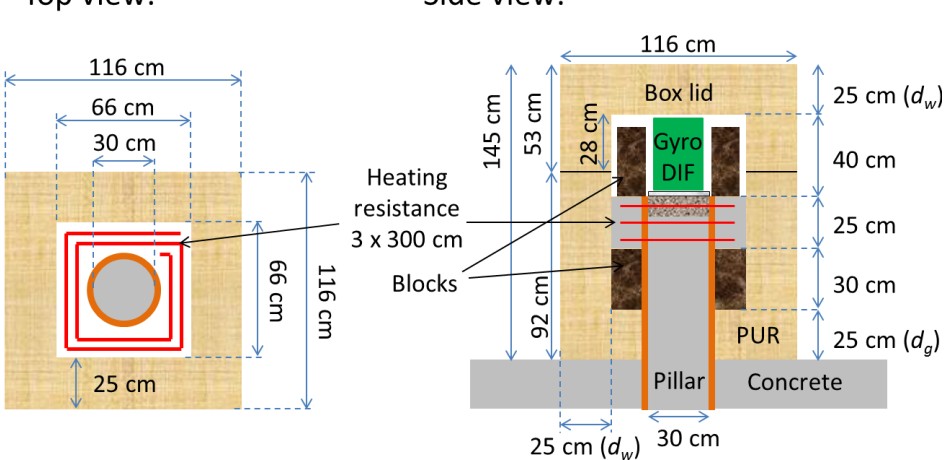

Figure 2: Layout of the GyroDIF box. The red spiral around the central pillar in the top view (left panel) represents the heating
resistance, which is arranged in three height levels as shown in the side view (right panel). The upper part of the box (above 92 cm
height) is a lid that allows access to the instrument.




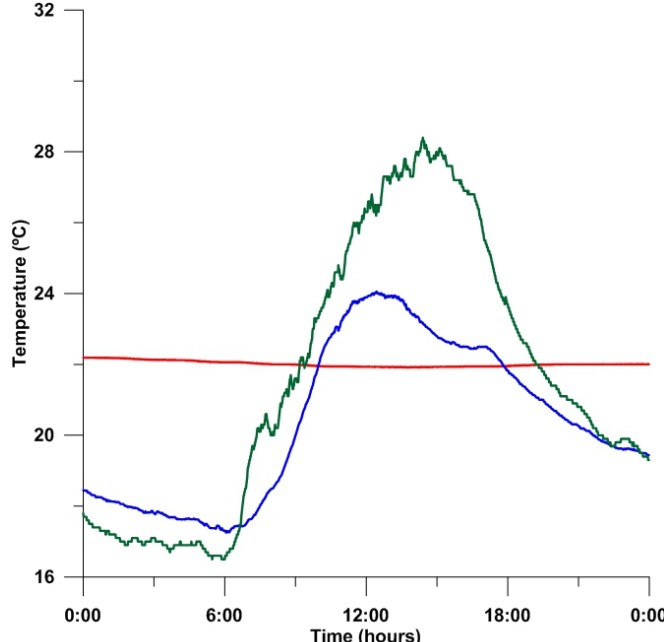

**Figure 3: Temperature variation within the box (red line), the igloo (blue line), and outdoor temperature (green line) during a test period at Ebre Observatory.**




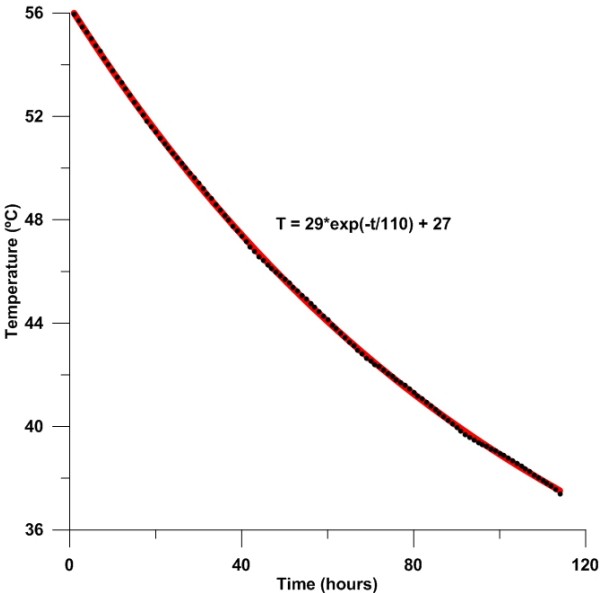

**Figure 4: Temperature decay after substantial heating of the box during a period of test at Ebre Observatory. The results from the exponential fit (red line) are shown in the inset: the most important parameter is the time constant $\tau = 110$ h (exponent in the inset equation), while the 29 and 27 parameter values (in ºC) depend on the specific experiment being performed.**

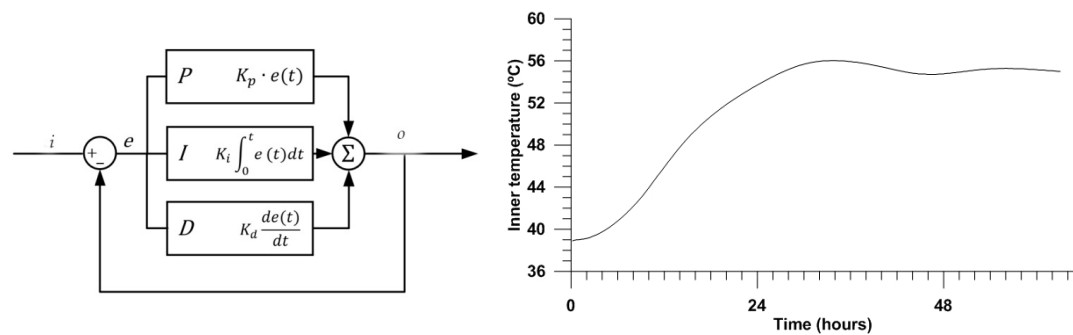

**Figure 5: (a) Scheme of a PID controller showing its three components: proportional (P), integral (I) and derivative (D). The error, $e$, is the difference between the desired value, $i$ (set point), and the achieved one, $o$. (b) Effect of the PID control on the box. Rise is observed up to the working temperature, followed by an oscillation which, in steady conditions, is about 0.3 ºC in a 1-day period**

10 **(test at Ebre Observatory).**

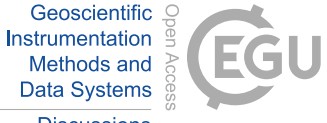



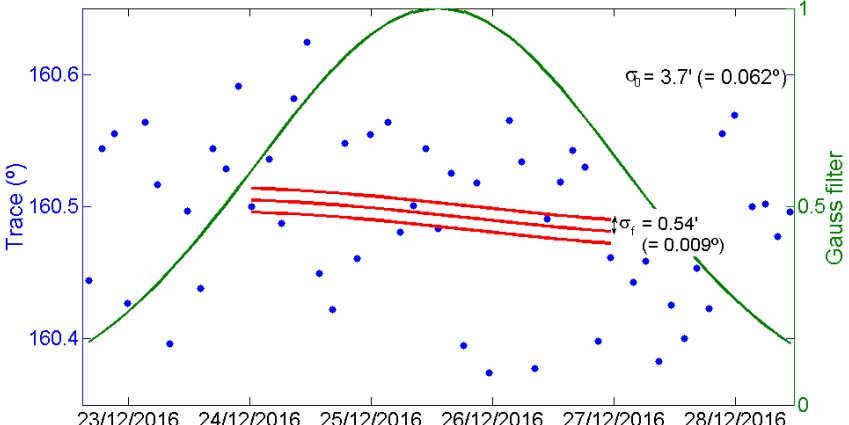

**Figure 6: Series of gyroscope True-North measurements (blue dots) at Dourbes observatory during a previous test period. The trace (left vertical axis) is the azimuth of the 0º mark of the theodolite's horizontal circle. Superposed (green line) is the Gaussian function (referred to the right vertical axis) filtering the observations, and the adopted baseline (red line; referred to the left vertical axis) along with its associated uncertainty.**





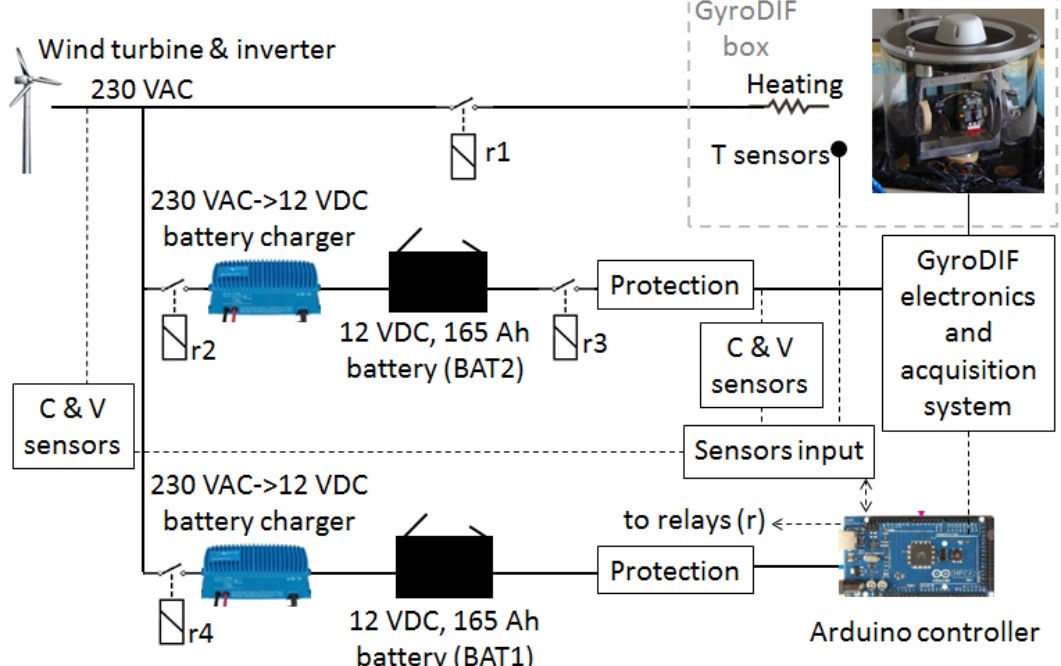

**Figure 7: Simplified electronic system layout. 230 VAC from wind generators feed the GyroDIF heating resistance and two batteries (BAT1 and BAT2) by means of their respective chargers. BAT2 in turn feeds the GyroDIF theodolite, its electronics and its acquisition system; BAT1 feeds an Arduino controlling the passage of current through the different parts of the system by acting on different switching relays (r) from the input given by a series of current and voltage (C & V) and temperature sensors.**