# Peer review of "An automatic DI-flux at the Livingston Island Geomagnetic Observatory, Antarctica: requirements and lessons learnt"

_Geoscientific Instrumentation, Methods and Data Systems, 2017_

## Referee Comment (RC1) · Anonymous Referee #1 · 19 Mar 2017

Review of Marsal et al., 'An automatic DI-flux at the Livingston Island Geomagnetic Observatory, Antarctica: requirements and lessons learnt'.

The manuscript describes the infrastructure and methods necessary to operate a GyroDIF, an automatic absolute instrument for geomagnetic measurements, at a semi-permanently occupied Antarctic station. Operational conditions for the AutoDIF and the GyroDIF are discussed.

The manuscript is very well organised and the quality of the figures is good or very good. The English can be improved. Regarding the figures, I suggest to improve figures 3, 4, 5b by adding axes on the upper and right side of the plot and generally to make them appear more aesthetic.

[Figure]

There are two shortcomings, and (minor) revision of the article would be an excellent opportunity to address them: Despite the title, we learn very little about the actual observations in Antarctica and any lessons learned there. Please be more informative about the current state of the instrument in Antarctica and possibly include additional data on the performance that was collected since the article was first submitted. Secondly, the list of references looks rather short. However, some concepts that are presented in the manuscript might have been used by observatory operations before and it might be valuable to mention where such concepts have been used before, or to cite relevant literature and/or review articles

Declination should be written as declination, not with capital D.

In the following, please find detailed comments. Most points regard the English language. Please check the whole manuscript for English language. p. stands for page l. stands for line, then comes the text which I find sounds strange. This is followed by a '->' symbol followed by suggestions for improvements or a question mark. ANY COMMENTS AND QUESTIONS ARE WRITTEN IN CAPITALS.

Title learnt -> learned (LEARNT IS OK, BUT LEARNED would be correct both in British and American English)

p.1

l. 9 magnetometer bar -> magnetometer

l. 12 practically removed -> ?

l. 15 due to its simpler necessary -> due to the simpler infrastructure that is necessary

l. 21 current society -> modern society

l. 23 augmentation of the -> improvement of

l. 26 UNCLEAR: do you mean these things have been automated or should be automated?

l. 28 learnt -> learned (IF YOU WANT)

p. 2

l. 5 RECORDS OF WHAT?

l. 6 measuring -> magnetic

l. 13 do no rely on a -> are not fixed to a

l. 28 rate-gyroscope -> ???

l. 30 custom electronic -> custom-made electronic

l. 31 govern -> control

p. 3

l. 1 leading to -> from the

l. 7 between the above -> either

l. 21 70 W. -> 70 W for the GyroDIF.

l. 23 path -> line of sight

l. 29 entailed -> impose

p. 4 l. 2 target -> desired

l. 6 the gyroscope -> the number of gyroscope

l. 10 no -> the absence of

l. 12 preserving in the system batteries in prevention of -> in order to overcome

l. 13 generation. -> generation by means of batteries.

l. 18 QUESTION: When you write masonry blocks, what do you specifically mean? Which type, what are they made of? Are they non-magnetic?

l. 23 QUESTION: foam glass, which type, is it non-magnetic?

p. 5

l. 12 winter -> local winter OR austral winter (PLEASE USE THIS ALWAYS WHEN YOU MENTION SUMMER OR WINTER)

l. 14 use to -> usually

p. 6

l. 30 CAN YOU RECOMMEND A FILTER?

p. 7

l. 16 the opposite is left to the natural cooling -> but cooling is passive.

p. 7

l. 18 WHY DO YOU CALL THIS BASELINE? IN MY VIEW, BASELINES ARE OFFESTS (THAT ARE SOMETIMES OBTAINED BY FILTERING MEASUREMENTS). HERE, THE MEASUREMENTS ARE FILTERED; BUT THIS DOES NOT LEAD TO A BASELINE IN THE CLASSICAL SENSE.

p. 8

l. 3 I WOULD COMPARE THE 3 nT ACCURACY WITH THE ACCURACY THAT IS NECESSARY FOR GETTING USEFUL SECULAR VARIATION DATA, AND NOT WITH THE 5 nT STANDARD BY INTERMAGNET.

p. 9

l. 14

perdurable -> ?

l. 14 generators from -> generators at

l. 26 system, which is duplicated -> control system, which is redundant

l. 28 passage of current through -> distribution of current to

p. 10 l. 3 next -> 2017????

Table 1 Comparative table -> Tale comparing

Figure 4 box around formula for T (you call it inset)

Figure 5b What is the set point used here?

Figure 6 Figure caption: I would not use the term 'baseline'

---

## Author Comment (AC1) · 29 Mar 2017

Response to reviewer #1

Find below the original reviewer's comments followed by our answers. I have also attached a modified version of the maniscript as a .pdf supplement, according to the reviewer's comments:

I suggest to improve figures 3, 4, 5b by adding axes on the upper and right side of the plot and generally to make them appear more aesthetic.

Done

[Figure]

There are two shortcomings, and (minor) revision of the article would be an excellent opportunity to address them: Despite the title, we learn very little about the actual observations in Antarctica and any lessons learned there. Please be more informative about the current state of the instrument in Antarctica and possibly include additional data on the performance that was collected since the article was first submitted. Information on the current state of the instrument has been updated. Secondly, the list of references looks rather short. However, some concepts that are presented in the manuscript might have been used by observatory operations before and it might be valuable to mention where such concepts have been used before, or to cite relevant literature and/or review articles.

We have added more literature.

Declination should be written as declination, not with capital D.

Done

Title learnt -> learned (LEARNT IS OK, BUT LEARNED would be correct both in British and American English).

Done

p.1 l. 9 magnetometer bar -> magnetometer

Done

l. 12 practically removed -> ?

We don't understand what the problem is with this sentence. The original sentence reads "practically reduced", not "practically removed" as the reviewer seems to suggest. In case our original sentence was wrong in English, we have replaced it with "practically restricted".

l. 15 due to its simpler necessary -> due to the simpler infrastructure that is necessary

Done

l. 21 current society -> modern society

Done

l. 23 augmentation of the -> improvement of

Done

l. 26 UNCLEAR: do you mean these things have been automated or should be auto-mated?

They should.

l. 28 learnt -> learned (IF YOU WANT)

We have changed it.

p. 2 l. 5 RECORDS OF WHAT?

Magnetic field records. This has been added in the manuscript.

l. 6 measuring -> magnetic

Done

l. 13 do no rely on a -> are not fixed to a

Done

l. 28 rate-gyroscope -> ???

Yes, this is correct. These gyroscopes measure rate of change of an angle, rather than a direction. We have added two references.

l. 30 custom electronic -> custom-made electronic

Done

l. 31 govern -> control

Done

p. 3 l. 1 leading to -> from the

Done

l. 7 between the above -> either

Done

l. 21 70 W. -> 70 W for the GyroDIF.

Done

l. 23 path -> line of sight

Done

l. 29 entailed -> impose

Done

l. 4 l. 2 target -> desired

Done

l. 6 the gyroscope -> the number of gyroscope

Done

l. 10 no -> the absence of

Done

l. 12 preserving in the system batteries in prevention of -> in order to overcome

Done

l. 13 generation. -> generation by means of batteries.

We have removed the part mentioning the batteries, since it may lead to confusion.

l. 18 QUESTION: When you write masonry blocks, what do you specifically mean? Which type, what are they made of? Are they non-magnetic?

We have added some clarification on this a few lines below in the text. No special condition applies to these blocks except non-magnetic properties. We just seek thermal momentum, implying a material with a high specific heat capacity. Solid stones of the type typically used in masonry are suitable because they are dense and easily stackable, thus resulting in a high overall heat capacity.

l. 23 QUESTION: foam glass, which type, is it non-magnetic?

Foam glass is a light insulating material made of glass and air bubbles also used in masonry. You can find more information on the particular mark we have used in http://www.foamglas.com/. Yes, it must be (and indeed it is) non-magnetic (we have specified it in the text).

p. 5 l. 12 winter -> local winter OR austral winter (PLEASE USE THIS ALWAYS WHEN YOU MENTION SUMMER OR WINTER)

Done. We have added "local" or "austral" in most occurrences of "winter" and "summer" when they were necessary.

l. 14 use to -> usually

Done

p. 6 l. 30 CAN YOU RECOMMEND A FILTER?

A moving average over a time window of 10 min is enough for our purposes. This has been added in the text.

p. 7 l. 16 the opposite is left to the natural cooling -> but cooling is passive.

Done

l. 18 WHY DO YOU CALL THIS BASELINE? IN MY VIEW, BASELINES ARE OFFESTS (THAT ARE SOMETIMES OBTAINED BY FILTERING MEASUREMENTS). HERE, THE MEASUREMENTS ARE FILTERED; BUT THIS DOES NOT LEAD TO A BASELINE IN THE CLASSICAL SENSE.

That's right. We have changed the text accordingly. However, we have kept the comparison with the baselines to note the resemblance of the filtering part.

p. 8 l. 3 I WOULD COMPARE THE 3 nT ACCURACY WITH THE ACCURACY THAT IS NECESSARY FOR GETTING USEFUL SECULAR VARIATION DATA, AND NOT WITH THE 5 nT STANDARD BY INTERMAGNET.

We think that comparing our expected accuracy with the INTERMAGNET standard is meaningful and worth because it is a reference for most geomagnetic observatories, but we have added a comparison with the accuracy necessary for secular variation studies, as suggested by the reviewer.

p. 9 l. 14 perdurable -> ?

Sorry, this is a Spanish/Catalan reminiscence. We have replaced it with durable.

l. 14 generators from -> generators at

Done

l. 26 system, which is duplicated -> control system, which is redundant

Done

l. 28 passage of current through -> distribution of current to

Done

p. 10 l. 3 next -> 2017????

Done

Table 1 Comparative table -> Tale comparing

Done

Figure 4 box around formula for T (you call it inset)

Done

Figure 5b What is the set point used here?

55 °C. It has been added in the caption.

Please also note the supplement to this comment:
http://www.geosci-instrum-method-data-syst-discuss.net/gi-2017-22/gi-2017-22-AC1-supplement.pdf

[Figure]

**Supplement:**

[revised manuscript text omitted]
 baselines of the magnetic field components in that a function  is often adjusted to a cloud of observations. Thus, given the substantial random uncertainty of the individual gyro determinations (around $\sigma_0 = 3.6'$; see Sect. 2), there is need to either filter or fit the observations to a known function so as to allocate a single value of the True North reference to each magnetic absolute measurement. We have currently opted for a Gaussian filter, i.e., the convolution of a Gaussian function with the True North observations, so that the maximum weight for a magnetic absolute determination at time $t_0$ is conferred to contemporary gyro measurements, while it is gradually reduced as the time shift increases.

[revised manuscript text omitted]

A robust electronic  control system, which is  redundant in some parts, has been designed to face the adverse conditions of the austral winter season, when the ASJI is unmanned. The intelligent Arduino-based control manages the distribution  of current to the different parts of the system in terms of power availability, and it integrates a PID algorithm adjusting the temperature of the GyroDIF box.

The necessary infrastructure of the new GyroDIF system has been successfully installed during the last austral summer survey, i.e., between December 2016 and February 2017. The installation of the instrument itself, however, has not been completed successfully due to the combination of a minor technical problem residing in a wire junction, the difficult logistics in Antarctica, and insufficient time for testing, but it is expected to be
10  left running unmanned during the  2018 austral winter season and beyond.

**Data availability**

No datasets have been used in the production of this article, except those experimental data obtained by the authors to generate figures 3, 4, 5b and 6.

**Author contribution**

15  S. Marsal designed part of the infrastructure containing the instrument, helped assembling and installing it in Antarctica, leads the data processing, and prepared the manuscript with contributions from all co-authors. J. J. Curto helped designing part of the infrastructure and electronics. J. M. Torta and J. J. Curto are the Principal Investigators of this project. A. Gonsette and J. Rasson are the instrument manufacturers. M. Ibañez, V. Favà and O. Cid helped designing the electronics controlling the instrument. M. Ibañez has leaded the installation of the necessary infrastructure in Antarctica.

20  **Competing interests**

The authors declare that they have no conflict of interest.

**Acknowledgments**

The presented results are part of the project CTM2014-52182-C3-1-P funded by the Spanish Ministerio de Economía y Competitividad. We are also grateful to the project 2016-URL-Trac-001 of the Universitat Ramon Llull. SM acknowledges
25  the support of Universitat Ramon Llull through project 2016-URL-IR2Q-029 funded by "Obra Social la Caixa".

**Tables:**

| | AutoDIF | GyroDIF |
|---|---|---|
|  Declination (and $Y$ component uncertainty | < 0.3′ ($\delta Y$ < 2 nT at LIV) limited by the laser pointing procedure | 1 gyro sequence ⇒ 3.6′ ($\delta Y \approx$ 19 nT). Quasi-continuous mode: $\delta Y \lesssim$ 3 nT |
| Power consumption | $\approx$ 20 W in average (total). | Heating: $\approx$ 12 W (average). Instrumentation and energy management: $\approx$ 60 W (average). Total: > 70 W |
| Necessary infrastructure | Complex: pipes or buried infrastructure. | Simple: thermally insulated box. |

Table 1. Table comparative ing AutoDIF vs. GyroDIF in terms of suitability at the ASJI.

**Figures:**

[Figure]

Figure 1: Left: Fiberglass igloo and insulating box (interior) for the GyroDIF thermal insulation at LIV. Right: GyroDIF box with
lid open inside the igloo; the GyroDIF theodolite is visible in the centre, along with the insulation, the masonry blocks, the
temperature sensors and the end of the heating cable.

[Figure]

Figure 2: Layout of the GyroDIF box. The red spiral around the central pillar in the top view (left panel) represents the heating
resistance, which is arranged in three height levels as shown in the side view (right panel). The upper part of the box (above 92 cm
height) is a lid that allows access to the instrument.

[Figure]

[Figure]

**Figure 3: Temperature variation within the box (red line), the igloo (blue line), and outdoor temperature (green line) during a test period at Ebre Observatory.**

[Figure]

**Figure 4: Temperature decay after substantial heating of the box during a period of test at Ebre Observatory. The results from the exponential fit (red line) are shown in the inset: the most important parameter is the time constant $\tau = 110$ h (exponent in the inset equation), while the 29 and 27 parameter values (in ºC) depend on the specific experiment being performed.**

[Figure]

**Figure 5: (a) Scheme of a PID controller showing its three components: proportional (P), integral (I) and derivative (D). The error, *e*, is the difference between the desired value, *i* (set point), and the achieved one, *o*. (b) Effect of the PID control on the box. Rise is observed up to the working temperature (set point of 55 ºC in this case), followed by an oscillation which, in steady conditions, is about 0.3 ºC in a 1-day period (test at Ebre Observatory).**

[Figure]

**Figure 6: Series of gyroscope True-North measurements (blue dots) at Dourbes observatory during a previous test period. The trace (left vertical axis) is the azimuth of the 0° mark of the theodolite's horizontal circle. Superposed (green line) is the Gaussian function (referred to the right vertical axis) filtering the observations, and the adopted  trace (red line; referred to the left vertical axis) along with its associated uncertainty.**

[Figure]

**Figure 7: Simplified electronic system layout. 230 VAC from wind generators feed the GyroDIF heating resistance and two batteries (BAT1 and BAT2) by means of their respective chargers. BAT2 in turn feeds the GyroDIF theodolite, its electronics and its acquisition system; BAT1 feeds an Arduino controlling the passage of current through the different parts of the system by acting on different switching relays (r) from the input given by a series of current and voltage (C & V) and temperature sensors.**

---

## Referee Comment (RC2) · C. Turbitt (Referee) · 24 May 2017

**Reviewer comments to gi-2017-22-manuscript-version v1**

The considered submission presents the results of initial tests of automated instruments capable of determining the magnetic vector in the geographic reference frame. Traditionally, these measurements have been made manually by skilled observers; however this requirement provides a limitation on the location and/or accuracy of absolute magnetic observatories. Here, the authors seek a robust and practical solution for the Spanish Antarctic Station on Livingston Island, which operates in a harsh environment and is unmanned through the winter months. The authors' findings provide are clear, well presented, thorough and provide a useful insight to some of the issues and potential resolutions encountered in operating remote, unmanned observatories that will be of interest to the wider geomagnetic observatory community.

The reviewer's recommendation is that the submission is published, on the proviso that the authors consider the following comments:

Page 2 line 7, "Variometric" is not a recognised term in English and either "variation magnetometers" or "variometers" would be preferred, although a GEM Systems proton magnetometer dos not fit either of these categories.

Page 2 line 13, the stable reference frame referred to here is the geographic reference frame and should be stated as such.

Page 2 line 16, the authors may wish to expand on what is meant by "baseline evolution" here to underline the problem to be solved i.e. what are the causes of variometer baseline evolution, what are typical signals (period, amplitude & resolution) requiring modelling by the baselines, why is a linear interpolation between occupations inadequate and what are the sampling recommendations of the international community?

Page 2 line 28, reference Rasson et Gonsette, 2016 here for the GyroDIF.

Page2 line 31, "In-house testing" is ambiguous. Does this refer to testing by the authors or a specification provided by the manufacturer? Also, which instrument does this refer to and is this consistent with the declination uncertainty figures in Section 2?

Page 3 line 9, do the authors have a reference for the assertion that the AutoDIF declination uncertainty is less than 0.3'? Similarly for the GyroDIF value of 3.6' stated later in the paragraph and in Table 1?

Page 4 line 26, it would be helpful to state here something about the driving current for the heating system as it is assumed that this must be safe and have no effect on the magnetometer enclosed within it.

Page 8 line 7, $\sigma_t$ should read $\sigma_f$

Suggested grammatical corrections:

Page 1 line 29, "..as it is _at_ our partially manned station.."

Page 2 line 23, "The one with longer history is called AutoDIF … designed to reproduce its manual measurement sequence", would read better as, "The one with the longest history is the AutoDIF … designed to reproduce the manual measurement sequence of the DI-flux."

Page 4 line 12, "preserving it in the system batteries in prevention of periods of scarce wind generation", would be better phrased as, "enabling its operation from system batteries during extended periods without wind-generated power."

Further notes:

Page 5 line 1, "basically" would be better replaced with "mostly". Note: heat loss by air exchange, which is easily estimated, can also be a lesser but significant factor as it is difficult to construct an air-tight enclosure. This may account for some or most of $P_a$. Some air exchange may be desirable e.g. to prevent the accumulation of damp.

---

## Author Comment (AC2) · 5 Jun 2017

Response to reviewer #2

Find below the original reviewer's comments followed by our answers. I have also attached a modified version of the maniscript as a .pdf supplement, according to both the reviewer #1 and reviewer #2 comments:

Page 2 line 7, "Variometric" is not a recognised term in English and either "variation magnetometers" or "variometers" would be preferred, although a GEM Systems proton magnetometer dos not fit either of these categories.

[Figure]

We have replaced "variometric magnetometers" with "variometers". We have also re-organized the succeeding text where we referred to the proton magnetometer as a variometer.

Page 2 line 13, the stable reference frame referred to here is the geographic reference frame and should be stated as such.

Done

Page 2 line 16, the authors may wish to expand on what is meant by "baseline evolution" here to underline the problem to be solved i.e. what are the causes of variometer baseline evolution, what are typical signals (period, amplitude & resolution) requiring modelling by the baselines, why is a linear interpolation between occupations inadequate and what are the sampling recommendations of the international community?

Done. We have expanded moderately our explanation on some aspects of our baseline at LIV, though we do not think we should go into much more detail here, mainly because this is not the main topic of the article and because we are in the Introduction section.

Page 2 line 28, reference Rasson et Gonsette, 2016 here for the GyroDIF.

Done

Page2 line 31, "In-house testing" is ambiguous. Does this refer to testing by the authors or a specification provided by the manufacturer? Also, which instrument does this refer to and is this consistent with the declination uncertainty figures in Section 2?

Even if we have preserved this sentence according to the original article by Hrvoic and Newitt (2011), we have explained what it means. In fact, this is a specification by the manufacturers referring to the AutoDIF. Also, the referee is right with the observation that the declination uncertainty stated here should be consistent with that given in Section 2. Indeed, both figures are not fully consistent, though they are reasonably close together: in this section we talk about 0.1', while in Section 2 we talk about "below 0.3' ". To avoid confusion we have rearranged the numeric figures somewhat:

note that 0.1' is the value given by the manufacturers in optimal conditions, though in the same article by Hrvoic and Newitt (2011) it is mentioned that some tests showed that the real accuracy was 0.2'. On the other hand, the value of < 0.3' stated in Section 2 was theoretically estimated by myself (the main author) on the basis of some coarse numbers (note that we are interested in an estimation of the uncertainty rather than a precise value of it), so we have replaced the sentence "below 0.3' " with "typically 0.2' ". We have also modified Table 1 accordingly. The conclusion is that the current AutoDIF accuracy is more or less 0.2', depending on the particular conditions of each observatory.

Page 3 line 9, do the authors have a reference for the assertion that the AutoDIF declination uncertainty is less than 0.3'? Similarly for the GyroDIF value of 3.6' stated later in the paragraph and in Table 1?

The < 0.3' figure was roughly estimated by the main author of the present article by considering the distance between the AutoDIF and its laser reflector in a preliminary project proposal where we were considering the most suitable option to us: either the AutoDIF or the GyroDIF. As for the stated GyroDIF value of 3.6', it is also an estimation by myself, but in this case the calculation is much more complex and would deserve a thorough theoretical development to justify it. It accounts for the noise of the gyro output signal and the particular procedure of the true-north angle measurement. Unfortunately, we can't give a reference for these figures, though we have tried to give some clues in the text.

Page 4 line 26, it would be helpful to state here something about the driving current for the heating system as it is assumed that this must be safe and have no effect on the magnetometer enclosed within it.

Done

Page 8 line 7, $\sigma$t should read $\sigma$f

[Figure]

Right. Done.

Suggested grammatical corrections: Page 1 line 29, "..as it is at our partially manned station.."

Done

Page 2 line 23, "The one with longer history is called AutoDIF . . . designed to reproduce its manual measurement sequence", would read better as, "The one with the longest history is the AutoDIF . . . designed to reproduce the manual measurement sequence of the DI-flux."

Great. Done.

Page 4 line 12, "preserving it in the system batteries in prevention of periods of scarce wind generation", would be better phrased as, "enabling its operation from system batteries during extended periods without wind-generated power."

Done

Further notes: Page 5 line 1, "basically" would be better replaced with "mostly". Note: heat loss by air exchange, which is easily estimated, can also be a lesser but significant factor as it is difficult to construct an air-tight enclosure. This may account for some or most of Pa. Some air exchange may be desirable e.g. to prevent the accumulation of damp.

Done. The referee is right with his observation that Pa partly accounts for heat loss by air exchange. However, we prefer to minimize it (and ideally prevent it) because the air humidity at LIV is usually very high, so new air coming in would imply more damp. Instead, we opted for an enclosure as airtight as possible containing a desiccant agent.

Please also note the supplement to this comment:
http://www.geosci-instrum-method-data-syst-discuss.net/gi-2017-22/gi-2017-22-AC2-supplement.pdf

**Supplement:**

**An automatic DI-flux at the Livingston Island Geomagnetic Observatory, Antarctica: requirements and lessons learnt**

Santiago Marsal[1], Juan José Curto[1], Joan Miquel Torta[1], Alexandre Gonsette[2], Vicent Favà[3], Jean Rasson[2], Miquel Ibañez[1], Òscar Cid[1]

[1]Observatori de l'Ebre, (OE), CSIC – Universitat Ramon Llull, 43520 Roquetes, Spain.
[2]Institute Royal Météorologique (IRM), Centre de Physique du Globe, 5670 Viroinval (Dourbes), Belgium.
[3]Institut de l'Ebre, 43500 Tortosa, Spain.

*Correspondence to*: Santiago Marsal (smarsal@obsebre.es)

**Abstract.** The DI-flux, consisting of a fluxgate magnetometer  coupled with a theodolite, is used for the absolute manual measurement of the magnetic field angles in most ground-based observatories world-wide. Commercial solutions for an automated DI-flux have recently been developed by the Royal Meteorological Institute of Belgium (RMI), and are practically restricted to the AutoDIF and its variant, the GyroDIF. In this article, we analyse the pros and cons of both instruments in terms of its suitability for installation at the partially manned geomagnetic observatory of Livingston Island, Antarctica. We conclude that the GyroDIF, even if less accurate and more power demanding, is more suitable than the AutoDIF for harsh conditions due to its simpler  infrastructure that is necessary. Power constraints in the Spanish Antarctic Station during the unmanned season impose an energy-efficient design of the thermally regulated box housing the instrument, as well as a thorough power management. Our experiences can benefit the geomagnetic community, who often faces similar challenges.

**1 Introduction**

Ground-based geomagnetic field data are currently used in a number of scientific works, from the Earth's deep interior to Space Weather studies, the latter with clear practical implications on our  modern society. As a consequence of these facts, the developments in instrumentation, data acquisition and data dissemination have increased the interest of the scientific community, and one of the most challenging aspects is the  improvement of the data coverage on remote sites such as oceanic regions in general, and the Southern Hemisphere in particular. For both logistical and economic reasons, full automation of the data acquisition is desirable, especially at those remote sites. There are several elements of geomagnetic observatory operation which should be fully or partially automated: data collection, data telemetry, data processing, data dissemination, error detection and absolute observations (Newitt, 2007). This paper is aimed at shedding some light on the practical aspects of the automation of absolute observations. We will report on the lessons learnt from the installation of an automatic absolute magnetometer in a particularly adverse environment, as it is at our partially manned station in Antarctica.

The Livingston Island geomagnetic observatory (62.7° S, 60.4° W, coded as LIV by the International Association of Geomagnetism and Aeronomy) is located in the Spanish Antarctic Station Juan Carlos I (ASJI), in the South Shetlands, north of the Antarctic Peninsula. Its first installation took place during the 1995–1996 and 1996–1997 Antarctic Surveys and it has

5 magnetic field records since December 1996. This observatory is manned during the austral summer months, typically from the end of November to February, being in automatic operation without human intervention the rest of the year. In terms of  magnetic instruments, it currently consists of  two variometers: a proton vector magnetometer in dIdD configuration and an FGE triaxial fluxgate magnetometer. As for absolute instruments, it has a DI-flux consisting of a Carl Zeiss THEO 015B theodolite equipped with an 810 Elsec

10 Fluxgate probe and  two GEM Systems proton magnetometers in different locations. For a detailed description of these instruments and the utility of the data provided by them see Pijoan et al. (2014) and references therein. For the purposes of this paper, it is interesting to note that variometers in general are automatic instruments with relatively high resolution, especially the combination FGE–scalar magnetometer, but they  are not fixed to the geographic reference frame. The DI-flux, on the contrary, is based on the geographic reference frame, but its measurements

15 have a lower resolution and, most importantly, they are necessarily manual. Thus, even if a tight absolute control is carried out during the survey months, the lack of absolute measurements during nine months a year when the station is unmanned prevents us from establishing reliable baselines for reduction of our variometer data. Note that the INTErnational Real-Time MAGnetic observatory NETwork (INTERMAGNET) recommends carrying out absolute measurements at least on a weekly basis (INTERMAGNET, 2012). The year-to-year baseline variation at LIV is moderate, typically below 2 nT/year, which

20 may justify our current assumption of a simple linear interpolation of the baselines between consecutive surveys; however,  we don't really know what the baseline evolution is during the winter months, when variations of the baselines arising from the different temperature conditions (with respect to the summer months) might be significant.

25 During the last 20 years we have progressed in practically all of the aforementioned aspects concerning the automation of LIV geomagnetic observatory. However, as for any institute that runs remote magnetic observatories, the automation of the absolute observations is of particular importance and the most challenging item, especially when the station is unmanned most of the time. At present, there have been very few attempts to automate absolute observations (Auster et al., 2007, 2009; Hrvoic and Newitt, 2011). The one with the longest history is  the AutoDIF (Rasson and Gonsette, 2011; Gonsette and

30 Rasson, 2013), an automat instrument designed to reproduce  the manual measurement sequence of the DI-flux. For what concerns the Declination measurement, the telescope of its theodolite is replaced by a laser and split photo cells which are used to align the device in a known meridian by reflecting the laser beam off a corner cube reflector back onto the photo cell. A recent variant is being provided that substitutes the target pointing system by an embedded device by which the True North referencing is achieved by a fibre optic rate-gyroscope (e.g., Pavlath, 1994; Gonsette et al.,

2017) able to detect the Earth rotation. This variant is accordingly called GyroDIF (Rasson and Gonsette, 2016). Both AutoDIF and GyroDIF use non-magnetic piezoelectric motors to move the sensor about the horizontal and vertical axes. The angles are measured by custom-made electronic optical encoders. An electronic bubble level mounted on the alidade provides reference to the horizontal. An embedded fan-less PC and a microcontroller  control the instrument. In-house testing by the manufacturers in optimal conditions has shown that the  AutoDIF can achieve an angular accuracy of 0.1′ (arc-minute), though some tests have revealed that this figure might be somewhat higher in real conditions.  This is comparable with the  accuracy that can be obtained by a skilled observer with a DI-flux. Although this instrument has given results that agree closely with those obtained by manual observations, long-term reliability under adverse conditions must yet be demonstrated (Hrvoic and Newitt, 2011).

In this article, we show our recent experience  from the installation of a GyroDIF at the ASJI in January–February 2017. This comprises the choice of the most suitable automatic absolute instrument based on the particular conditions in our station, as well as the design of the necessary infrastructure to accommodate it. Because the instrument deployment has not culminated during the last austral summer survey  our experience is limited, so in this text we will combine in situ LIV data with real data taken in Ebre Observatory headquarters and data at the manufacturer's site (Dourbes observatory) during  periods of test in 2016.

**2. AutoDIF vs. GyroDIF**

First, we had to assess the most suitable option, either  the AutoDIF or the GyroDIF. Assuming mechanical stability of the pillars where the variometers are deployed, just a few absolute magnetic determinations per week are required. The oretical estimations show that the uncertainty of a single declination measurement with the AutoDIF is typically  0.2′ if the laser reflector is far enough, which is roughly translated into 1 nT in the magnetic east component ($Y$) at LIV. On the other hand, the north-seeking gyroscope procedure of the GyroDIF, which is used for reference in the Declination measurements, has an expected standard deviation $\sigma_0$ around 3.6′, (according to the specifications of the noise in the gyro output and the particular measurement procedure). This  translates into an uncertainty of 19 nT in $Y$ at LIV. Fortunately, however, the dispersion of the latter measurements shows a white noise signature, allowing it to be overcome by a sufficiently large number of them. In continuous operation, the GyroDIF uncertainty can thus be substantially reduced.

Another factor to be considered for our choice was power consumption, since the ASJI relies on wind generators during the austral winter. The total average power (including idle and active periods) required for the AutoDIF plus the power needed to keep the instrument above 5 ℃ (its minimum working temperature) is probably less than 15–20 W. For the GyroDIF operating in a continuous mode, however, we need a constant temperature, demanding below about 15 W for the heating if

we are capable of providing a good thermal insulation. Adding the power for the instrumentation, as well as that for management of the station, this value raises to more than 70 W for the GyroDIF.

Thirdly, we had to assess the necessary infrastructure for each kind of equipment. The AutoDIF requires a clear line of sight for the laser beam between the instrument and the reflector target which is difficult to get in the Antarctic environment because of weather conditions resulting in reduced visibility (snow, fog, rain …). The reflector needs to be separated by at least 30 m from the instrument, though preferably 100 m. A first possibility would be to underground the installation. However, the difficult terrain and the fact that Livingston Island is a protected environment hamper this option. The second possibility was to build a pipe visually connecting the AutoDIF and the reflector, but the strong winds and the instability of the terrain again imposed technical difficulties. The GyroDIF option, on the other hand, is simpler: we just needed to provide a highly insulated box containing the instrument. Therefore, even if less accurate and more power demanding, the GyroDIF seemed the best option concerning both logistics and stability, and we finally opted for it.

Table 1 summarizes the main differences between the AutoDIF and the GyroDIF from the point of view of its suitability at the  desired site.

**3. Installation requirements**

Let us analyze the GyroDIF specific requirements in more detail:

- To avoid damage in the piezo-motors acting on the horizontal and vertical axes, the instrument must work above 5 ℃.

- We need to maximize the number of gyroscope True North sampling s so as to reduce the random uncertainty of the Declination observations.

- The gyro response and, in consequence, the uncertainty in the magnetic $Y$ component, also depends on the temperature variation during a single measurement, so thermal stability must be guaranteed.

- The gyroscope response is also sensitive to external accelerations; in consequence, the instrument location requires the absence of motion by wind, sea waves or others, which introduce additional noise into the True North measurements.

- Finally, it is essential to minimize power consumption,  enabling its operation from system batteries during extended periods without wind-generated power.

**3.1 Thermal model of the GyroDIF box**

[revised manuscript text omitted]

A robust electronic control system, which is  redundant in some parts, has been designed to face the adverse conditions of the austral winter season, when the ASJI is unmanned. The intelligent Arduino-based control manages the distribution  of current to the different parts of the system in terms of power availability, and it integrates a PID algorithm adjusting the temperature of the GyroDIF box.

The necessary infrastructure of the new GyroDIF system has been successfully installed during the last austral summer survey, i.e., between December 2016 and February 2017. The installation of the instrument itself, however, has not been completed successfully due to the combination of a minor technical problem residing in a wire junction, the difficult logistics in Antarctica, and insufficient time for testing, and it is expected to be left running unmanned during the  2018 austral winter season and beyond.

**Data availability**

No datasets have been used in the production of this article, except those experimental data obtained by the authors to generate figures 3, 4, 5b and 6.

**Author contribution**

S. Marsal designed part of the infrastructure containing the instrument, helped assembling and installing it in Antarctica, leads the data processing, and prepared the manuscript with contributions from all co-authors. J. J. Curto helped designing part of the infrastructure and electronics. J. M. Torta and J. J. Curto are the Principal Investigators of this project. A.

Gonsette and J. Rasson are the instrument manufacturers. M. Ibañez, V. Favà and O. Cid helped designing the electronics controlling the instrument. M. Ibañez has leaded the installation of the necessary infrastructure in Antarctica.

**Competing interests**

The authors declare that they have no conflict of interest.

**Acknowledgments**

The presented results are part of the project CTM2014-52182-C3-1-P funded by the Spanish Ministerio de Economía y Competitividad. We are also grateful to the project 2016-URL-Trac-001 of the Universitat Ramon Llull. SM acknowledges the support of Universitat Ramon Llull through project 2016-URL-IR2Q-029 funded by "Obra Social la Caixa".

**Tables:**

| | AutoDIF | GyroDIF |
|---|---|---|
| -Declination (and $Y$ component uncertainty | 0.2′ ($\delta Y \simeq$ 1 nT at LIV) limited by the laser pointing procedure | 1 gyro sequence $\Rightarrow$ 3.6′ ($\delta Y \approx$ 19 nT). Quasi-continuous mode: $\delta Y \lesssim$ 3 nT |
| Power consumption | $\approx$ 20 W in average (total). | Heating: $\approx$ 12 W (average). Instrumentation and energy management: $\approx$ 60 W (average). Total: > 70 W |
| Necessary infrastructure | Complex: pipes or buried infrastructure. | Simple: thermally insulated box. |

Table 1. Table Comparing AutoDIF vs. GyroDIF in terms of suitability at the ASJI.

**Figures:**

[Figure]

Figure 1: Left: Fiberglass igloo and insulating box (interior) for the GyroDIF thermal insulation at LIV. Right: GyroDIF box with lid open inside the igloo; the GyroDIF theodolite is visible in the centre, along with the insulation, the masonry blocks, the temperature sensors and the end of the heating cable.

[Figure]

Figure 2: Layout of the GyroDIF box. The red spiral around the central pillar in the top view (left panel) represents the heating resistance, which is arranged in three height levels as shown in the side view (right panel). The upper part of the box (above 92 cm height) is a lid that allows access to the instrument.

[Figure]

[Figure]

**Figure 3: Temperature variation within the box (red line), the igloo (blue line), and outdoor temperature (green line) during a test period at Ebre Observatory.**

[Figure]

**Figure 4: Temperature decay after substantial heating of the box during a period of test at Ebre Observatory. The results from the exponential fit (red line) are shown in the inset: the most important parameter is the time constant $\tau$ = 110 h (exponent in the inset equation), while the 29 and 27 parameter values (in ºC) depend on the specific experiment being performed.**

[Figure]

**Figure 5: (a) Scheme of a PID controller showing its three components: proportional (P), integral (I) and derivative (D). The error, $e$, is the difference between the desired value, $i$ (set point), and the achieved one, $o$. (b) Effect of the PID control on the box. Rise is observed up to the working temperature (set point of 55 ºC in this case), followed by an oscillation which, in steady conditions, is about 0.3 ºC in a 1-day period (test at Ebre Observatory).**

[Figure]

**Figure 6: Series of gyroscope True-North measurements (blue dots) at Dourbes observatory during a previous test period. The trace (left vertical axis) is the azimuth of the 0º mark of the theodolite's horizontal circle. Superposed (green line) is the Gaussian function (referred to the right vertical axis) filtering the observations, and the adopted  trace (red line; referred to the left vertical axis) along with its associated uncertainty.**

[Figure]

**Figure 7: Simplified electronic system layout. 230 VAC from wind generators feed the GyroDIF heating resistance and two batteries (BAT1 and BAT2) by means of their respective chargers. BAT2 in turn feeds the GyroDIF theodolite, its electronics and its acquisition system; BAT1 feeds an Arduino controlling the passage of current through the different parts of the system by acting on different switching relays (r) from the input given by a series of current and voltage (C & V) and temperature sensors.**